# Using Evidence-Based Practice and Data-Based Decision Making in Inclusive Education

Gabrielle Wilcox *, Cristina Fernandez Conde and Amy Kowbel

School and Applied Child Psychology Program, Werklund School of Education, University of Calgary, Calgary, AB T2N 1N4, Canada; cristina.fernandezco@ucalgary.ca (C.F.C.); amy.kowbel@ucalgary.ca (A.K.)
* Correspondence: gwilcox@ucalgary.ca

**Abstract:** There are longstanding calls for inclusive education for all regardless of student need or teacher capacity to meet those needs. Unfortunately, there are little empirical data to support full inclusion for all students and even less information on the role of data-based decision making in inclusive education specifically, even though there is extensive research on the effectiveness of data-based decision making. In this article, we reviewed what data-based decision making is and its role in education, the current state of evidence related to inclusive education, and how data-based decision making can be used to support decisions for students with reading disabilities and those with intellectual disabilities transitioning to adulthood. What is known about evidence-based practices in supporting reading and transition are reviewed in relationship to the realities of implementing these practices in inclusive education settings. Finally, implications for using data-based decisions in inclusive settings are discussed.

**Keywords:** inclusive education; data-based decision making; transition planning; reading disabilities; intellectual disabilities

## 1. Using Evidence-Based Practice and Data-Based Decision Making in Inclusive Education

Twenty-five years ago, Vaughn and Schumm [1] outlined the components of responsible inclusive education. These components included teacher choice and the development of their own philosophy of inclusion, adequate resources and professional development, a continuum of services rather than having only one option: full inclusion, school-based models rather than district- or state-mandated models, putting student needs first, ongoing evaluation of effectiveness, and curriculum and instructional practices that meet student needs. Making decisions that put students first ensures that students make progress academically and socially through ongoing progress monitoring. Decisions about placement and programming should be based on data related to student progress toward their goals rather than assuming that the same placement and programming will meet all students' needs. To understand some of the limitations in the use of data within the context of inclusive education, this article describes evidence-based practice and data-based decision making (DBDM) within the context of inclusive education, specifically in the areas of reading and transition to adulthood for students with intellectual disability (ID). In this article, we reviewed data-based decision making and how it is used to support educational outcomes, and the current state of evidence regarding the effectiveness of inclusive education. We then provided examples of how evidence-based practices related to DBDM can be used to make decisions in inclusive education that prioritizes meeting students' needs. We used two examples: reading difficulties and intellectual disability (ID) to provide examples with groups of students who have different needs.

## 2. Evidence-Based Practice and Data-Based Decision-Making

DBDM is a process of gathering data about how students are progressing toward specific goals in academic or behavioral performance. This includes identifying the current

and desired levels of performance, implementing an evidence-based intervention, regularly monitoring progress toward meeting that goal, and modifying the intervention as necessary [2]. This is an iterative process rather than a few steps to follow through once. DBDM is a process that can be implemented at all levels from entire districts to individual students. While staff at the school district and individual school levels should use data to inform their decisions about how they educate and support the academic and social-emotional development of their students, this article will focus on DBDM as it relates to individual students with exceptional needs.

While data are an important component of DBDM, decision makers must *interpret* the data to inform decisions about how to effectively support students. Data must be combined with pedagogical and content knowledge to translate it into a usable action plan, taking the context into consideration [3]. This reasoning process is not as straightforward as it appears, and decision makers need to attend to potential cognitive biases and misapplied heuristics that can interfere with their decision making, for example, confirmation bias [4,5]. While evidence suggests that DBDM can improve student outcomes [6], more work is needed to effectively translate this into widespread practice. Unfortunately, although teachers collect a great deal of data, they rarely use it explicitly in decision making for individual students' progress [7].

The outcomes of decisions and interventions implemented within the classroom depend on the validity of the inferences drawn from the data. Unfortunately, data literacy tends to be low among school personnel, contributing to this limited use of data in decision making [8,9]; however, supporting staff understanding of data can increase their data literacy [10]. Data literacy concerns the ability to analyse and interpret data so that it can be used to inform practice. Data only becomes usable information when the observer is able to understand it, which involves multiple steps. First, it is necessary to collect and organize data related to the goal and then transform the data into information. Then, educators need to be proficient in analyzing and summarizing, creating concise and targeted summaries of relevant information. Finally, by synthesizing the information into a unified and usable summary and prioritizing what has the most importance in working toward the intended goals, the information becomes useable knowledge. Teachers can use this knowledge to determine the effectiveness of an intervention, creating a feedback loop to the previous stages which informs changes needed to increase intervention effectiveness [3]. The complexity of this process may lead to false interpretations if the educator is not proficient in these skills.

Formal professional development in areas of data use can be difficult to access. Often, knowledgeable staff members train principals or other administrative staff, who then train teachers, relying on colleagues rather than development programs [9,11]. This training tends to be brief, without consistent levels of quality, and with a focus on using data systems rather than on data interpretation or how to connect the resulting information with strategies for instructional improvement [9]. Even within schools that promote and support the use of data, data are rarely used to improve teaching or adapt instruction to meet the needs of students [9,12]. Most often teachers respond by looking at the content of instruction, re-teaching or retesting the relevant information, or forming groups based on the identified needs of students rather than adjusting the delivery methods of their instruction [12]. To appropriately use the data collected in the classroom, teachers must be given the opportunity to improve their skills in data literacy. Means and colleagues [9] suggest that collaboration can be useful in this process, allowing teachers to learn from each other, clarify any problems, correct errors, and bring a broader range of interpretive skills to the task. Greater access to technological resources such as student information systems, instructional management systems, assessment systems, and diagnostic systems, can also help teachers meaningfully collect, analyse, and communicate data when the complexity becomes overwhelming [11]. Although DBDM is a process that can support effective teaching and positive student outcomes, teachers receive little to no training in this area and often have limited time to devote to data analysis and interpretation.

### 3. Evidence and Data in Inclusive Education

The rhetoric of inclusivity has made it unpopular to suggest that evidence of its effectiveness is necessary to support its implementation [13]. However, we argue with Reynolds and colleagues [14] (p. 307), that while "in God we trust. All others must have data." Ensuring that students receive effective instruction and support is of the utmost importance, and we cannot support a claim without evidence. While inclusive education is a popular idea, teachers generally do not use evidence-based curricula let alone evidence-based interventions for students with special education needs (SENs) [15]. Additionally, in inclusive settings, teachers report spending less time with students who have SENs [16], making it unlikely that they are effectively using student data to make decisions about instruction and intervention to ensure positive academic and social and emotional outcomes for students. Meta-analyses have found mixed results of the impact of inclusive education. They also noted limitations of studies examining inclusive education outcomes that included differing definitions of inclusion, amount of inclusion, varying levels of student needs included in the studies, and a lack of control groups [17,18]. Furthermore, a review evaluating the evidence supporting inclusive education indicates that there is no clear evidence regarding the positive effects of inclusion [19].

One challenge with some conceptions of inclusive education is with the idea that all students, no matter the level of need or level of education (from primary to high school) should be fully included in regular education classes in order to protect human rights [20,21]. However, both teacher training and the level of student need impact teacher readiness to provide education and whether or not teachers leave the field. In a study of teacher turnover, Gilmour and colleagues [22] found that regardless of their training, teachers who taught students with behavioral challenges left the field at higher rates. Teachers with special education degrees had lower turnover when working with students diagnosed with specific learning disabilities, often in special education settings, while those with dual degrees had higher turnover when they had students with higher needs, and there was a higher rate of teacher turnover in regular education classes with higher numbers of students with SENs.

Relatedly, there is a belief that labelling is stigmatizing. While labelling can be used in a way that is stigmatizing [23], it can also be empowering, explaining difficulties so that students do not just feel "stupid." For example, one woman shared her story of being identified with a learning disability as an adult and how this helped to her get the interventions she needed to learn to read and have an explanation for her difficulties [24]. Furthermore, obtaining diagnostic clarity can be an advantage. The mother of a young boy with autism noted that confirming the diagnosis helped her understand her son's strengths and areas of need [25]. In addition, full inclusion for all does not take student or parent preferences into account, as found in one study in which parents whose children were in special education classrooms (92%) were more pleased with their children's placement than those who were placed into inclusive classrooms (47%). Moreover, more students were happy with their placement in a special education program (92%) than in mainstream classrooms (64%) in a small study in the UK [26].

While teachers regularly collect information on student progress, they are less likely to use that information to inform their teaching strategies or to inform programming decisions for students with SENs, despite the fact that students with severe disabilities perform better when teachers follow guidelines for DBDM. Teachers often note that they do not have the training necessary to interpret student data especially when there are multiple sources of data that add to the complexity of analysis [9,27]. Many teachers struggle to make sense of data representations, differences or trends, and creating relevant goals related to the data, which can lead to invalid inferences [9]. When data are complex, teachers can lose track of their initial goals or begin to rely on their own impressions and past experiences rather than the data [9]. Additionally, a qualitative study of primary school teachers in Belgium found that teachers reported using intuition rather than data to determine whether or

not to retain students [28] despite the strong evidence that retention does not improve long-term academic outcomes for students (see [29]).

Teachers should monitor the progress of students with SENs toward their educational goals through Individual Education Plans (IEPs), which are legal documents that are also tools for data-based decision making. A Turkish study examining the IEPs of students with SEN in inclusive settings found that only one third of the legal requirements were met on average in these IEPs [30]. For example, less than half included students' current level of performance and personal information, and when they included that information, it was vague. A majority of the IEPs did not contain information about special education services or related services. These are necessary components of DBDM, suggesting that there is not enough information in those IEPs to make data-based decisions about student progress. Unfortunately, research collecting data on inclusive education often focus on teacher and pre-service reflections rather than student outcomes [31,32]. Currently, there is limited research on the effectiveness of inclusive education, and the research that exists has significant limitations and has demonstrated mixed findings. There is also limited research on the use of DBDM in inclusive settings, but research on teaching practices [16], attrition [22], and use of IEPs [30], which document DBDM, suggest that it is not used effectively in inclusive education settings.

## 4. How Data Can Be Used to Inform Inclusive Education in Practice and Research?

This section highlights two common diagnoses of school-aged children with different aetiologies and different needs to demonstrate the role of DBDM for students with SEN. Reading difficulties are common. In the United States, just over a third meet proficiency [2]. IDs are less prevalent with 1–3% of the population having this diagnosis, and they tend to have more intensive needs related to functional academics and skills of daily living [33]. Consequently, these two groups of students provide a wide view of the types of services students require and how DBDM can be implemented in inclusive settings.

## 5. Data-Based Decision Making in Reading

Elementary school teachers should use evidence-based curriculum and curriculum-based measurement (CBM) to determine if students are gaining reading skills at an adequate rate because early intervention is critical for students' long-term outcomes. Students who do not gain reading skills early are less likely to ever catch up, especially if they do not receive remediation. They are also likely to experience multiple negative outcomes including less educational attainment, limiting their career choices; greater risk of drop out and interaction with the criminal justice system; and increased rates of mental health diagnoses [34–36]. Additionally, reading difficulties are quite common with 42% of Canadians not possessing the reading skills necessary to follow multi-step directions and 25% of third graders not reading independently [37,38]. They are costly, not only to the students who do not attain adequate reading skills but also to the global economy, costing over USD one trillion every year [39,40].

If inclusive education is considered to be a continuum where students spend as much time as possible in regular education classrooms as long as their needs can be met, then many students with reading challenges should be able to be included in regular education for most or all of the school day if DBDM is used to ensure that the reading interventions they are receiving are helping them to reach their reading goals. There is a strong foundation of research regarding best practices in reading instruction and intervention that could be used to support inclusive education for students with reading disabilities including how to collect and use data to make decisions about adequate progress [39]. Effective reading instruction depends on gathering and using reliable data to identify individual student needs and areas of strength [41]. Teachers then need to modify instruction or intervention strategies to improve reading outcomes if students are not making adequate progress to reach their reading goals [42]. Teachers need preservice and in-service training in delivering

evidence-based general curriculum as well as training in CBM in order to engage in the DBDM necessary to meet the needs of students with SEN.

Areas of need in reading are initially identified through universal screeners that evaluate overall school performance and determine which students need additional supports to successfully gain reading skills [43]. CBM is used to monitor growth in a specific skill to evaluate whether instruction is effective or not [44]. It is important to train teachers and staff who will be administering the measures to safeguard the fidelity and reliability of the data [44]. Similarly, it is crucial to monitor student motivation and to find a quiet distraction-free setting for test administration [44]. Research on the accuracy of decision making demonstrates the importance of gathering accurate data and to avoid making decisions solely based on judgment or intuition [45]. When decision making teams use multiple pieces of data (i.e., universal screening data, CBM, in-class assessments), they increase decision accuracy [45]. To use DBDM to support reading effectively, teachers need to collect and interpret the data and use it to inform instruction. However, even when teachers can read and interpret data and progress monitoring graphs, it is still difficult for them to link the data to instruction [46]. Linking data to instruction is more challenging than just reading data. Thus, specific training to link data to instruction is necessary for teachers to do so effectively [46]. There is a strong foundation of research on effective reading instruction, intervention, and CBM that could be used to support inclusive education practices for students with reading difficulties.

## 6. Realities of Delivering Inclusive Reading Education

While teachers who understand reading development and evidence-based reading instruction are better able to choose appropriate interventions for struggling students, teachers generally do not receive adequate training in this area, and experience does not improve their understanding of how to teach reading [47]. For example, in an international study of professionals in reading education, only 40% indicated that teacher training programs provide adequate training in reading instruction [48]. If teachers do not have the training to implement evidence-based reading instruction, which is a general education strategy, it is unlikely that they have the skills necessary to use data to determine specific skills to support struggling students and to select and implement more intensive interventions.

One study found that barriers to using CBM to support literacy skills included limited coaching support, not knowing how to translate assessment data into support for students, limited teacher knowledge, and reluctance to examine teaching practices as a result of student data [49]. Additionally, teachers do not receive adequate training in how to use data to make instructional decisions. For example, Wagner and colleagues [50] examined pre-service teachers' skill in reading and interpreting graphs of student CBM data related to their progress during a reading intervention. They found that pre-service teachers not only said less about what CBM graphs indicated than experts, they also provided descriptions that demonstrated weak understanding of the sequence of components of CBM. This is not surprising as teachers report receiving little if any training in CBM [51]. This is particularly concerning in an inclusive education setting where teachers are expected to meet the needs of students with a variety of challenges in academic skill acquisition in addition to other areas of need.

Reading disabilities are an area with significant research and clear guidelines on how to use data to make decisions and determine appropriate interventions. This research has demonstrated that an effective reading intervention requires intensive interventions that are delivered frequently, in small groups, over several months using explicit direct instruction with high levels of student engagement and practice [52]. These are skills that elementary teachers in inclusive settings need in order to support students with reading difficulties in gaining the reading skills necessary for academic success. Additionally, inclusive education teachers need to be skilled in collecting data on student progress towards their goals, in making decisions as to whether or not students are making adequate progress toward their goals, and how to modify the interventions when they are not. Unfortunately, students

with disabilities require more intensive instruction, and their low achievement can often be attributed to them not receiving instruction at the required intensity in regular education classrooms [53]. In order to demonstrate that inclusive education can meet the needs of the high incidence reading disabilities, teachers need to have knowledge of reading development and instruction as well as how to collect and interpret data on student progress so that they can use that data to make decisions about student progress and modify interventions as needed. Additionally, teachers need time and flexibility to deliver small group and one-on-one intensive interventions to students who are struggling while meeting the needs of other students in their class.

## 7. Data-Based Decision Making in Transition Planning Students with IDs

Best practices in transition planning for students with intellectual disabilities (IDs) need to be based upon DBDM. These include observable and measurable postsecondary goals that are monitored regularly to inform educational changes that need to be made to support reaching those goals [54]. Evidence-based practices to support transition to adulthood include instruction and support in job training, work study, life skills training, self-determination training, functional academics, parents involved in the transition planning process, social skills training related to workplace relationships, and community agency collaboration [55,56]. Other secondary school experiences related to better postsecondary outcomes for students with disabilities generally include work experience, parental involvement, independent living skills, social skills, vocational education, work study, parent expectations, youth decision making, travel skills, and goal setting [57,58]. While a meta-analysis noted that students with disabilities performed better when involved in inclusive classrooms, few studies were included, and these had variable sample sizes, with small percentages of students with IDs (e.g., 12.38%) [57]. Consequently, broad studies of students with disabilities are not likely to adequately represent the experiences of students with ID.

There are many evidence-based interventions that target self-determination, which is important to support outcomes for students with ID. Self-determination concerns students' agency over their lives and education, making choices for themselves, and taking initiative in creating and attaining their goals [59]. Increasing self-determination can also improve decision making, goal setting, problem solving, self-monitoring and self-regulation [60]. Students with ID who scored higher in self-determination were more likely to live independently three years after high school, demonstrate greater financial independence, have positive employment outcomes, and have better community access [61,62]. A randomized placebo-controlled trial demonstrated that interventions improved self-determination in students with ID [63].

Data on students' current level of performance and their postsecondary goals are necessary to create measurable, observable transition goals in order to identify the transition planning for individual students' needs and to inform DBDM [64]. It is important to accurately communicate what will be expected of the student, the gap that currently exists, and what remediation and accommodations will be used to support students in reaching those goals, which will also be used to measure progress [64]. Though future-oriented goals are essential to the transition process, many teachers feel uncomfortable about helping students develop goals related to adulthood, which they cannot measure at the endpoint [65]. Because of this discomfort, teachers struggled to specify postsecondary goals and to develop clear and concise annual goals and how they would be measured [65]. Unfortunately, without clear and measurable annual goals, it is difficult to collect relevant data, monitor progress, or evaluate how to support the student, which leads to poor outcomes after they transition to adulthood [66]. Research on effective transition to adulthood for ID clearly outlines the experiences that increase their successful transition [54–58]. These areas are often difficult to provide in inclusive high school settings, however, because job training and independent living skills are typically not covered in preparing students for university.

## 8. Realities of Delivering Inclusive Education for Students with ID

When students graduate from high school, they are expected to be prepared for postsecondary training or to begin contributing to society through employment or volunteer experiences and to have meaningful community connections. To highlight the reality that full inclusion is not universally beneficial, we summarize the findings from Hornby and Kidd [26]. In the UK, some students with mild ID were transferred to full-time regular education classrooms as part of a policy to increase inclusive education. After they graduated, 71% of those who transferred to regular education were unemployed; 67% still lived with their parents; 46% indicated that they had no friends, 25% had only one friend. When they were asked about their most useful school experience, only 13% reported their time in regular education, while 46% reported that their time in special education was the most useful. Although this was a small study and the quality of instruction in the various settings was not accounted for, it suggests that inclusive education placement is not universally beneficial for students with ID.

Much of the research on inclusion for students with ID focuses on social benefits of being in an inclusive classroom, but it fails to take into account that the ideals of inclusivity are not always the reality, and some students with disabilities actually choose to socially interact with others who have similar needs and are more accepting of them [67]. Additionally, parents can have different inclusion goals for their children. In one small study, a mother wished for more inclusion, while the other wanted more special education to develop independent living skills. Neither student received the services their mothers thought they needed, contributing to challenges during the transition process [68]. These studies highlight the unique needs and goals of students that cannot all be met within the same setting and require a flexible continuum of services to provide students with the education they need to meet their postsecondary goals.

Research supporting inclusive education often only measures socialization [69], but long-term outcomes for students with ID are often predicated on educational experiences that support vocational and independent living skills. The areas of skill development that support effective transition to adulthood include training in independent living skills, self-determination, social skills and work experience [55–57]. There are, however, few if any high school teachers who can effectively teach the curriculum of their subject area and adequately support students with ID to gain these additional skills, especially in secondary education [70]. One qualitative study investigated parent and teacher perceptions of how the transition of students with ID is supported within an inclusive setting [71]. Most parents described their experience with inclusive education negatively, noting that their children's needs for instruction in basic skills necessary for academic success were largely ignored, resulting in a widening of the achievement gap between them and their peers [71]. Parents also reported that their children did not receive instruction in independent living, social, and travel skills within mainstream education. Consequently, parents needed to find these supports on their own. Though some teachers strive to learn more about evidence-based practices, there is a large gap between best practices and actual practice in transition planning [72]. In order to implement best practices in transition, teachers need to have knowledge and skill in delivering and measuring the effectiveness of interventions and the space in their curriculum to deliver them [72].

Though evidence suggests that transition goals related to independent living and vocational skills have a significant effect on improving the skills and outcomes of students with ID, they are rarely implemented in schools [61]. This could be a result of a lack of training in areas of evidence-based practices in transition. The majority of teachers and transition professionals report having little or no professional development or training in transition evidence-based practices, and even those who did receive some level of professional development reported that it did not prepare them to implement those practices [72,73]. In order to support students with ID as they transition into adulthood, it is important to provide them with training that will support them in gaining the independence needed to find success in adulthood.

## 9. Implications of the Lack of CBM and DBDM for Inclusive Education

Currently, teachers frequently report being unprepared to teach the wide range of student needs in inclusive settings, especially when students have behavioral challenges [74]. In order to meet the needs of students with SENs, teachers in inclusive settings need not only subject matter and pedagogical knowledge, they also need a wide range of instructional strategies and evidence-based interventions from which they can pull to meet their students' varying needs, because if students were able to be successful with general pedagogical knowledge, they would not need an IEP [22]. Relatedly, it is important to consider what is reasonable to expect from a teacher in how much they can reasonably differentiate instruction, especially considering the high rates of teacher burnout and turnover [70,75,76]. For example, it may be unreasonable to expect all teachers be skilled in supporting using picture exchange systems, toileting and feeding, and specific academic interventions in addition to classroom management, general instructional strategies, and their content [77].

Additionally, teachers need training in DBDM [10], including in collecting, interpreting, and using CBM and other progress monitoring tools to provide instruction [46]. Without preservice training or additional support, it is unlikely that DBDM will have a positive impact on instruction. Teachers need to be able to make decisions based on different sources of data as this relates to student progress toward their goals in order to prevent instructional decisions based solely on their judgement. As noted above, training is needed not only in data collection and interpretation, but it is also crucial for teachers to select and use appropriate progress monitoring tools. In addition to research on data-based decision making for students with disabilities broadly, we also need to increase a culture of using data to make decisions on instruction and placement for students with disabilities in order to engage in responsible inclusion and receive the type and intensity of instruction and feedback that they require to make progress toward their goals [53,70]. Without training in evidenced-based instructional practices, DBDM, and CBM, it will be challenging for teachers to know if students are making adequate progress or what additional supports or instructional modifications may be important to support students who are not making adequate progress.

Finally, inclusive education needs to prioritize student needs and use DBDM to ensure that students are receiving the interventions and supports necessary to reach these goals. For some students, this will mean full inclusion. Other students, however, will need access to services that are not available in inclusive settings whether that is a pull out program to provide an intensive reading intervention or a partial or full-time placement in a special education setting to provide instruction in independent living and vocational skills. As Zigmond and Kloo [78] noted, "general education is a place; special education is a service." (p. 161). In order to meet the needs of students with SENs, we need to ensure that they receive the services they need first, then consider where those services can be provided. These decisions are complex and require a full understanding of student and parent goals as well as the skill sets of teachers and supports available in schools. Vaughn's and Schumm's [1] call for responsible inclusion is still relevant, and those decisions require the use of DBDM to support *all* students in meeting their educational goals.

**Author Contributions:** Conceptualization: G.W.; writing—original draft: G.W., C.F.C. and A.K.; writing—review and editing: G.W., C.F.C. and A.K.; supervision: G.W. All authors have read and agreed to the published version of the manuscript.

**Funding:** This research received no external funding.

**Institutional Review Board Statement:** Not applicable.

**Informed Consent Statement:** Not applicable.

**Data Availability Statement:** Not applicable.

**Conflicts of Interest:** The authors declare no conflict of interest.

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
