# Peer review of "Using Evidence-Based Practice and Data-Based Decision Making in Inclusive Education"

_education, doi:10.3390/educsci11030129_

Round 1

Reviewer 1 Report

First of all, I would like to congratulate the authors for their article, the subject matter is interesting and inclusive education is always an important aspect of day-to-day life in schools and society.

This is a theoretical article, with no research methodology, and current and interesting references have been used.

Some time ago, this article would have been considered interesting, in the present times, even theoretical or reflective articles must, if they want to be published in impact journals, such as the one we are dealing with, be based on some minimum indicators of qualitative methodology, in particular.

Specifically, the sections:

Evidence-Based Practice and Data-Based Decision-Making
Evidence and Data in Inclusive Education
How Data Can be Used to Inform Inclusive Education in Practice and Research? 
Data-Based Decision Making in Reading
Realities of Delivering in Inclusive Reading Education
Data-Based Decision Making in Transition Planning Students with ID

Inclusive Education, then, why choose those sections and not others, why choose those sections and not others? 
what scientific basis do the authors have for this order and selection,
Why did they choose students with intellectual disabilities and not other disabilities?
Normally, the topics should have been chosen on the basis of a meta-analysis or a meta-analysis or a systematic review.

The section on the realities of inclusive education shows the same deficiency,
Where is the systematic review or meta-analysis that supports this section?

Similarly, if it is an article of reflection, there is a lack of basic authors, such as Ainscow and Booth, and more current ones
such as Arnaiz (2019, 2020), who rethinks the current direction of inclusion.

With all this, I conclude the non-acceptance of this article, as it is a review of some aspects of inclusion, not justified, as it is a review of some aspects of inclusion, not justified.
of inclusion, not justified, and without scientific basis in its preparation.

Author Response

Comment

Response

Inclusive Education, then, why choose those sections and not others, why choose those sections and not others? 
what scientific basis do the authors have for this order and selection,

The format of our manuscript is similar to another published manuscript in this issue. Kauffman & Hornby, 2020. The goal of this manuscript is to highlight the evidence for DBDM and to make the argument that it is currently not being used within inclusive education in order to have evidence of support students effectively.

Why did they choose students with intellectual disabilities and not other disabilities?

Added a sentence in the introduction (p. 2) paragraph describing why we chose these two groups (p.8) under “How data can be used to inform inclusive education in practice and research?”

Normally, the topics should have been chosen on the basis of a meta-analysis or a meta-analysis or a systematic review.

The format of our manuscript is similar to another published manuscript in this issue. Kauffman & Hornby, 2020

The section on the realities of inclusive education shows the same deficiency,
Where is the systematic review or meta-analysis that supports this section?

The format of our manuscript is similar to another published manuscript in this issue. Kauffman & Hornby, 2020

Similarly, if it is an article of reflection, there is a lack of basic authors, such as Ainscow and Booth, and more current ones
such as Arnaiz (2019, 2020), who rethinks the current direction of inclusion.

Added a sentence with a reference from Azorín & Ainscow and to Sánchez. We did not add more because the articles we read by these authors were not related to DBDM, which is the primary focus of this manuscript.

Reviewer 2 Report

The authors emphasize that current research is under the philosophy of inclusive education.

They highlight that there is very little information on the role of data-based decision-making in inclusive education.

The authors reviewed data-based decision-making and its role in education.

More specifically they explored and reflected on how data-based decision making can be used to support decisions for students with reading disabilities and those with intellectual disabilities.

To support their argument authors highlighted the usefulness of the data-based decision making indicating benefits explaining that it ensures that students make progress academically and socially through ongoing progress monitoring.

The authors pointed out the benefits of using  DBDM  in terms of identifying current and desired levels of performance, implementing an evidence-based intervention, regularly monitoring progress toward meeting that goal, and modifying the intervention as necessary.

Finally, the authors discussed and reflected implications for using data-based decisions in inclusive settings.

The authors clarified that Data must be combined with pedagogical and content knowledge to translate it into a usable action plan, taking the context into consideration.

In addition, they highlight the ethical issue of the reasoning process which is not as straightforward and decision-makers need to attend to potential cognitive biases and misapplied heuristics that can interfere with their decision making.

Authors suggest valuable implications highlighting that  In order to meet the needs of students with SEN, teachers in inclusive settings need not only subject matter and pedagogical knowledge, they also need a wide range of instructional strategies and evidence-based interventions from which they can pull to meet their students’ varying needs.

Authors could have highlighted beneficial aspects of DBDM in terms of psychological factors of inclusion in a separate chapter.

The authors should state more clearly the limitations of the suggested DBDM in a separate chapter.

Author Response

This is a summary of the manuscript; there were no comments regarding necessary changes.

Reviewer 3 Report

Reviewer report

Dear Author(s):

In accordance with the review of the article "Using Evidence-Based Practice and Data-Based Decision Making in Inclusive Education" (education- 1120621), which has been assigned to my person.

Synopsis of the review

In my opinion, the authors have done a really good job, but from my point of view, there are several flaws that do not encourage the publication of the work until they are solved. I explain:

- In my opinion, the paper needs to be restructured in a more efficient way in order to make it easier to read for a prospective reader. Obviously, this journal does not follow strict requirements in that sense but, according to the guide for authors, they should follow a structure as indicated:

"We do not have strict formatting requirements, but all manuscripts must contain the required sections: Author Information, Abstract, Keywords, Introduction, Materials & Methods, Results, Conclusions, Figures and Tables with Captions, (...)".

Therefore, I recommend that you adapt your format to these specifications. On the other hand, the inclusion of tables and/or figures is not an explicit obligation, but I also recommend that you add a "float" to summarize the content and, again, make the reading of a future reader more "friendly" than in its current state.

In the introduction, indicate in a summarized form the main topics covered in your article and what are the proposed achievements.

- Bibliographical references:

It is quite striking to note the citation "XXX" and the entry in the bibliographic references "XXX". I do not know what may be the motive, but it seems to me a nonsense. If it is to "hide" a self-citation, I understand that it is not the right way. In any case, remove the XXX and cite appropriately.

You should probably reconsider the title "Using Evidence-Based Practice and Data-Based Decision Making in Inclusive Education". You acknowledge that " there is little empirical data to support full inclusion for all students and even less information on the role of data-based decision making in inclusive education specifically". To my mind, this is somewhat contradictory. Have you considered including the word "Review" in your title.

For my part, once the recommended changes are made, I would be honored to have this paper accepted.

Best regards,

The reviewer.

Author Response

Comment

Response

- In my opinion, the paper needs to be restructured in a more efficient way in order to make it easier to read for a prospective reader. Obviously, this journal does not follow strict requirements in that sense but, according to the guide for authors, they should follow a structure as indicated:

We have added summaries with stronger transitions to improve the structure.

Therefore, I recommend that you adapt your format to these specifications. On the other hand, the inclusion of tables and/or figures is not an explicit obligation, but I also recommend that you add a "float" to summarize the content and, again, make the reading of a future reader more "friendly" than in its current state.

We have added a few summary sentences at the end of each section.

In the introduction, indicate in a summarized form the main topics covered in your article and what are the proposed achievements

We have added several sentences in the introduction outlining what is covered in this manuscript. (p. 2)

It is quite striking to note the citation "XXX" and the entry in the bibliographic references "XXX". I do not know what may be the motive, but it seems to me a nonsense. If it is to "hide" a self-citation, I understand that it is not the right way. In any case, remove the XXX and cite appropriately.

We did not do this; it must have been done by the editorial staff.

You should probably reconsider the title "Using Evidence-Based Practice and Data-Based Decision Making in Inclusive Education". You acknowledge that " there is little empirical data to support full inclusion for all students and even less information on the role of data-based decision making in inclusive education specifically". To my mind, this is somewhat contradictory. Have you considered including the word "Review" in your title.

While there is no evidence specific to inclusive education, we provide the evidence that is it helpful in improving outcomes in the populations we have highlighted. The manuscript highlights that this EBP is not implemented in inclusive education and how doing so would improve outcomes for students. 

There is no research on DBDM in inclusive education, so it is not a review. We note how DBDM, which is an expectation in special education could be applied in inclusive education to improve student outcomes.

Round 2

Reviewer 1 Report

First of all, I would like to congratulate the authors, I can see that the revisions, although complicated, have been carried out perfectly, giving a twist to the article and undoubtedly improving its quality, the bibliographic update has been key in this process, as well as the methodological revision, my congratulations.

Reviewer 3 Report

Dear author(s),

I consider that you have carried out the improvement proposals suggested by me (more or less), so in my opinion, this work should be accepted in its present state.

Best,

The reviewer.